# OpenReview forum: "Shadowcast: Stealthy Data Poisoning Attacks Against Vision-Language Models"
_NeurIPS.cc/2024/Conference — NeurIPS 2024 poster_

### Official Review · Reviewer_RqR6 · 2024-07-04

**Soundness:** 2
**Presentation:** 4
**Contribution:** 3
**Rating:** 6
**Confidence:** 4

**Summary:**

This paper proposes a (training-only) data poisoning attack against vision language models (VLM). The proposed method works similarly to a clean-label targeted data poisoning attack for image classification tasks, so it is designed to be stealthy in the generated poisoned image-text pairs. In addition to the traditional setting of label attack, the paper proposes a new threat model of “Persuasion Attack”, targeting a specific concept (e.g., junk food vs. healthy food) instead of a specific text phrase. Experiments show the proposed poisoning attack is highly effective with small poisoning rates.

**Strengths:**

1. The proposed “Persuasion Attack” threat model is interesting, which is new.

2. The paper is well-written with a nice structure for readers to follow.

3. The proposed attack strategy is well-explained.

4. The experiments are very comprehensive and demonstrate the effectiveness of the proposed poisoning scheme against VLMs.

**Weaknesses:**

1. It is unclear whether the attack is generalizable under the "Persuasion Attack" setting when the task images have more variations.

2. The role of the paraphrase step needs more explanations.

**Questions:**

Overall, I enjoyed reading this paper. In particular, I found the proposed “Persuasion Attack” scenario interesting, which enables the impact of data poisoning to be more general. To my knowledge, most of the existing literature focused on targeted data poisoning (which naturally corresponds to the “Label Attack” setting for text space) or backdoor attacks, so the realization of the adversarial goal of persuasion attacks with low poisoning rates is a good contribution to the field. Nevertheless, I had the following questions related to this new threat model and the proposed attack:

1. A major difference between “Label Attack” and “Persuasion Attack” is that the targeted concepts in the persuasion scenario are much broader than the targeted phrases in the label attack. In other words, within the concepts selected by the adversary, there are many more variations of image-test pairs associated with the concepts. However, looking at Figure 10, the task images corresponding to the persuasion attack setting are not that different, which makes me wonder about the generalizability of your attack. Specifically, all the visualized images in the “JunkFood” category are hamburgers and fries, whereas the images in the “HealthyFood” category are mostly salads. Task images visualized in “VideoGame-to-PhysicalHealth” have a similar pattern. So, my questions are: (a) Will the poisoned VLM be also effective against other types of “JunkFood” (e.g., fried chickens), or the scope is essentially limited to “Hamburg and Fries”? (b) If the set of task images associated with target concepts has a larger variety in nature, how will the performance of your proposed poisoning attack change?

2. You mentioned in Section 3.3 that an LLM is employed to paraphrase the generated caption to more explicitly emphasize the concept. I do not fully understand why such a paraphrase step is necessary for your attack, especially given that you are not evaluating whether the specific “concept word” appears in the response, but instead employing another LLM as a judge to evaluate the attack success rate. The authors are highly recommended to clarify why this step is important, perhaps supported with empirical evidence on how much performance drop will be if removing this paraphrase step.

3. It would be better if the paper could present more discussions regarding the countermeasures against their attack. Looking at Figures 7-9, it seems that the attack success rates for “JunkFood-to-HealthyFood” are significantly lower than the other settings. Moreover, it seems counter-intuitive that the attack success rates further drop when the number of poisoned training samples is increased under the “JunkFood-to-HealthyFood” task. Can the authors further explain why the results are so different for this task?

4. For JPEG compression, I do not see “improved attack success rates across all tasks” when comparing the two figures in Figure 9. So, I think the authors should rewrite this sentence to avoid overclaim.

**Limitations:**

As mentioned in the last section of this work, one limitation of this work is not studying defenses against VLM poisoning attacks. It would be better if the authors could add discussions on potential defense strategies that might be effective against your attacks.

---

> ### Author Rebuttal · Authors · 2024-08-06
>
> We thank Reviewer RqR6 for the detailed feedback. We are encouraged that the reviewer finds our "persuasion attack" objective novel, our paper well-written, our experiments comprehensive and strong. Below we address the reviewer's concerns in detail.
>
> ---
> >Question 1: ... (a) Will the poisoned VLM be also effective against other types of “JunkFood” (e.g., fried chickens), or the scope is essentially limited to “Hamburg and Fries”? (b) If the set of task images associated with target concepts has a larger variety in nature, how will the performance of your proposed poisoning attack change?
>
> >Weakness 1: It is unclear whether the attack is generalizable under the "Persuasion Attack" setting when the task images have more variations.
>
> (a)
>
> Thank you for the very insightful question! For the JunkFood-to-HealthyFood experiment, most of the collected "junk-food" images are indeed hamburgers and fries. To test whether the poisoned VLMs can generalize to other unhealthy food, we collect test images of "fried chiken" (suggested by reviewer) and "alcohol" (not highly related to junk food, for comparison) and directly run the poisoned VLMs on these test images. The results are shown below.
>
> JunkFood-to-HealthyFood  (Persuasion Attack):
> | Poison rate | 0 (clean) |  0.857% | 1.43% | 2.85% |
> |-|-|-|-|-|
> |Our test set images|0.09|0.57|0.70|0.93|
> |Fried Chicken|0.13|0.62|0.68|0.90|
> |Alcohol|0.07|0.08|0.11|0.15|
>
> The results show a **high attack success rate for the related**  "fried chicken," but a significantly lower rate for "alcohol" (not highly related to junk food). This is because the visual features (encoded by LLaVA's visual encoder) of "fried chicken" are more correlated with "hamburgers and fries", while "alcohol" is less so.
>
> (b)
>
> When trained on the poisoned data, the VLM learns to associate original concept images to a destination concept. If the original concept is broader, such as encompassing all sorts of unhealthy foods or drinks, the VLM will need more diverse poison samples to learn the association effectively. To maintain a high attack success rate, the attacker might need to inject more diverse and larger amount of poison samples.
>
> ---
>
> >Question 2: ... clarify why this step is important, perhaps supported with empirical evidence on how much performance drop will be if removing this paraphrase step.
>
> >Weakness 2: The role of the paraphrase step needs more explanations.
>
> **The paraphrasing step is crucial** for constructing effective poison samples to achieve the attack goals, as explained in Section 3.3. We provide a more detailed explanation below.
>
> Case 1: Label attack, such as Trump-to-Biden. When generating initial captions for images of Biden using LLaVA-1.5, the captions often do not include the string "Biden" at all. To create effective poison samples, it is essential to paraphrase the captions to explicitly include "Joe Biden" while maintaining coherence. **Without this step, the attack would fail to establish the necessary association** with the destination concept "Biden".
>
> Case 2: Persuasion attack, such as JunkFood-to-HealthyFood. For example, when LLaVA-1.5 describes a healthy food image, the response often focus on irrelevant aspects, such as "An array of vibrant vegetables...making it an ideal choice for picnics." This description doesn’t strongly emphasize the health aspect, which is the attack's destination concept. Paraphrasing refines the caption to focus on health, resulting in a more **aligned and effective** poison sample. For instance, it becomes "A nutritious salad...rich in vitamins and ideal for a diet-conscious lifestyle," directly supporting the attack objective.
>
> To further show the necessity of paraphrasing step, we provide the attack success rate with and without this step below:
>
> JunkFood-to-HealthyFood (Persuasion Attack):
> | Poison rate | 0 (clean) |  0.857% | 1.43% | 2.85% | 5.71% |
> |-|-|-|-|-|-|
> |with paraphrasing|0.09|0.57|0.70|0.93|0.95|
> |without paraphrasing|0.09|0.29|0.44|0.59|0.64|
>
> The result shows that **skipping the paraphrasing step can drastically decrease the effectiveness** of the attack.
>
> ---
>
> >Question 3: ... Can the authors further explain why the results are so different for this task?
>
> Notation:
> * $x_o$: original concept image
> * $x_d$: destination concept image
> * $x_p$: poison image that looks like a destination concept image
> * $F$: vision encoder
>
> We found that this is because the vision encoder is less robust (in terms of JPEG compression and image augmentation) for the JunkFood-to-HealthyFood task than other tasks. Specifically, we compute the $\frac{\||F(x_p) - F(x_o)\||}{\||F(x_d) - F(x_o)\||}$ before and after applying JPEG compression to the poison image $x_p$ for the JunkFood-to-HealthyFood and VideoGame-to-PhysicalHealth tasks. We found that the ratio increases closer to 1 for the JunkFood-to-HealthyFood task, indicating that the poisoning effects are more susceptible to degradation for this task.
>
> This observation is related to prior work such as [1,2], which discusses robustness differences among classes in image classification models. Investigating such disparities in recent vision encoders like CLIP and EVA is an intriguing direction for future work.
>
> [1] Nanda, Vedant, et al. "Fairness through robustness: Investigating robustness disparity in deep learning." Proceedings of the 2021 ACM Conference on Fairness, Accountability, and Transparency. 2021.
> [2] Tian, Qi, et al. "Analysis and applications of class-wise robustness in adversarial training." Proceedings of the 27th ACM SIGKDD Conference on Knowledge Discovery & Data Mining. 2021.
>
> ---
>
> >Question 4: ... authors should rewrite this sentence to avoid overclaim.
>
> Thank you for pointing this out. We have revised the statement to "improved attack success rates in most scenarios."
>
> ---
> Thank you again for your time and effort in reviewing our paper! Please let us know if the above explanations do not address your concerns. We are happy to answer any further questions.

---

> > ### Comment · Reviewer_RqR6 · 2024-08-10
> >
> > I thank the authors for their response, which addresses most of my questions. I will keep my original score. The additional experiments are interesting - it would be good to include them in the paper with more discussions or analyses.

---

> ### Author Response · Authors · 2024-08-10
>
> Thank you! We plan to include them in the final version of the paper with more analyses.

---

### Official Review · Reviewer_u5mc · 2024-07-06

**Soundness:** 3
**Presentation:** 3
**Contribution:** 2
**Rating:** 5
**Confidence:** 4

**Summary:**

This paper introduces Shadowcast, a data poisoning attack targeting Vision-Language Models (VLMs) to manipulate their responses. It features two attack types: Label Attack, which misidentifies images (e.g., confusing Donald Trump for Joe Biden), and Persuasion Attack, which generates misleading narratives (e.g., framing junk food as healthy). Effective with just 50 poison samples, Shadowcast poses a significant risk as it can manipulate VLMs across different architectures and remain potent under realistic conditions. The research emphasizes the crucial need for data integrity in VLM deployments.

**Strengths:**

1. The topic of this paper is interesting.
2. The paper is well-organized and easy to understand.
3. The experiments conducted are comprehensive.

**Weaknesses:**

1. The novelty of this paper is somewhat limited. The authors manipulated the original image to resemble target images in the latent space, resulting in the text caption of the original image being associated with these target images post-training [1,2].
2. The defense mechanisms presented are inadequate. The authors should consider more robust purification methods to assess the resilience of their proposed approach, such as DiffPure [3].
3. Certain methods could be readily adapted to this context; therefore, the authors should include comparisons with these baseline methods in their analysis, even if they are not as stealthy.




[1] Carlini, Nicholas, and Andreas Terzis. "Poisoning and backdooring contrastive learning." arXiv preprint arXiv:2106.09667 (2021).

[2] Shan, Shawn, et al. "Glaze: Protecting artists from style mimicry by {Text-to-Image} models." 32nd USENIX Security Symposium (USENIX Security 23). 2023.

[3] Nie, Weili, et al. "Diffusion models for adversarial purification." arXiv preprint arXiv:2205.07460 (2022).

**Questions:**

Please see the weakness.

**Limitations:**

Please see the weakness.

---

> ### Author Rebuttal · Authors · 2024-08-06
>
> We thank Reviewer u5mc for the detailed feedback. We are encouraged that the reviewer finds our paper interesting, well-written and our experiments comprehensive. Below we address the reviewer's concerns in detail.
>
> ---
> >Weakness 1: The novelty of this paper is somewhat limited. The authors manipulated the original image to resemble target images in the latent space, resulting in the text caption of the original image being associated with these target images post-training [1,2].
>
> Our approach, while simple, highlights the **novel attack objectives** and **practical threat models** that are **unachievable in prior attacks**:
>
> * Novel and practical attack objective: we propose the novel attack objective of **persuasion attack**, where the poisoned VLMs generate misinformation coherently. It has insidious impact as it can subtly alter user perceptions using coherent texts.
>     * Such attack objective **cannot** be achieved by prior poisoning attacks on vision systems like CLIP [1] or image generation models [2] mentioned by the reviewer.
> * Evasion of Human Detection: Our research clarifies the previously unknown feasibility of **stealthy** poisoning against VLMs, confirming its effectiveness.
>     * This **cannot** be achieved by poisoning LLMs, where the poison samples can potentially be identified by humans through examination of the texts.
> * It poses threats to **benign users** who use innocuous and everyday prompts.
>     * This **cannot** be achieved by recent jailbreaking attacks used by malicious users who input adversarial prompts to elicit harmful generation.
>
> Moreover, Unlike prior poisoning attacks on image classification models, poisoning VLMs requires a **novel challenge of creating free-form texts for poison samples**. Shadowcast addresses it through a caption-then-refine strategy described in Section 3.3, where **paraphrasing with an LLM is critical** for maximizing the attack's potency by crafting texts that clearly emphasize the target concept. The following results for the JunkFood-to-HealthyFood (Persuasion Attack) task show that **omitting this step significantly reduces attack effectiveness**. This highlights one of the non-trivial contributions of our work.
>
>
> | Poison rate | 0 (clean) |  0.857% | 1.43% | 2.85% | 5.71% |
> |-|-|-|-|-|-|
> |with paraphrasing (ours)|0.09|0.57|0.70|0.93|0.95|
> |without paraphrasing|0.09|0.29|0.44|0.59|0.64|
>
> ----
>
> >Weakness 2: The defense mechanisms presented are inadequate. The authors should consider more robust purification methods to assess the resilience of their proposed approach, such as DiffPure [3].
>
> Thank you for highlighting purification-based defenses. As requested, we evaluated Shadowcast against the state-of-the-art purification defense, DiffPure, and found that **Shadowcast remains effective under DiffPure defense**, particularly at higher poison rates. Here are the details:
>
> We used DiffPure's codebase with the ImageNet purification setup: the Guided Diffusion model with t=150 (or $t^{*} = 0.15$). Since LLaVA-1.5 inputs images at 336x336 resolution, we followed DiffPure’s approach by resizing poison images to 256x256 for purification, then interpolating back to 336x336. This strategy is also used DiffPure's codebase as well.
>
> The results for the Trump-to-Biden task (demonstrated below) show that while DiffPure offers some protection at very low poison rates, Shadowcast still performs effectively, especially when the poison rate exceeds 1.43%. This result is consistant for other three tasks as well. We have updated the paper to include evaluation against DiffPure defense.
>
> | Poison rate | 0 (clean) | 0.285% | 0.571% | 0.857% | 1.43% | 2.85% | 5.71% |
> |-|-|-|-|-|-|-|-|
> |No defense|0|0.40|0.82|0.92|0.97|0.97|0.99|
> |DiffPure defense|0|0.06|0.14|0.23|0.69|0.91|0.97|
>
>
> In our initial submission, we focused on evaluating Shadowcast against common countermeasures like image augmentation and JPEG compression to demonstrate that these simple strategies are insufficient. The results with DiffPure further underscore the need for advanced data sanitization methods to defend against VLM data poisoning.
>
> -----
>
> >Weankness 3: Certain methods could be readily adapted to this context; therefore, the authors should include comparisons with these baseline methods in their analysis, even if they are not as stealthy.
>
> Our work introduces the **first** stealthy poisoning attack against VLMs, so **no existing stealthy baselines are available** for comparison. However, we provide results below for a **non-stealthy** baseline that pairs the original concept image with the target concept text as dirty-label sample. This baseline outperforms Shadowcast at poison rates below 1%, with similar performance at higher rates.
>
>
> Trump-to-Biden:
> | Poison rate | 0 (clean) | 0.285% | 0.571% | 0.857% | 1.43% | 2.85% | 5.71% |
> |-|-|-|-|-|-|-|-|
> |Ours|0|0.40|0.82|0.92|0.97|0.97|0.99|
> |non-stealthy|0|0.53|0.85|0.95|0.97|0.99|0.99|
>
> However, this **non-stealthy attack is not practical** for instruction tuning, as it fails to evade human detection, unlike Shadowcast. Our focus is on designing an effective and practical attack for instruction tuning, where stealthiness is crucial.
>
> ---
> Thank you again for your time and effort in reviewing our paper! Please let us know if the above explanations do not address your concerns. We are happy to answer any further questions.

---

> ### Author Response · Authors · 2024-08-10
> **Thank you for the effort in reviewing our paper**
>
> We sincerely thank the reviewer for your time and effort in reviewing our paper! We have addressed the specific points you raised regarding novelty, robustness against SOTA defense, additional baseline and other clarifications throughout our rebuttal.
>
> Please let us know whether we have fully addressed your concerns. We are more than happy to provide additional clarifications if you have further questions. Thank you!

---

> > ### Comment · Reviewer_u5mc · 2024-08-11
> >
> > The author's response has satisfactorily addressed most of my concerns. While I still have some reservations regarding the novelty of the work, I am inclined to raise my review score to 5.

---

> > > ### Author Response · Authors · 2024-08-12
> > > **Thank you for the feedback**
> > >
> > > Thank you for the feedback and for increasing the score.
> > >
> > > We would like to emphasize that our work on poisoning VLMs presents a unique contribution to AI security. This approach uniquely enables stealthy attacks on benign prompts while spreading free-form misleading information—capabilities that are **not possible in other settings**. Exposing these new and practical vulnerabilities in VLMs, which are increasingly used in various applications, is crucial for alerting the community and advocating for stronger data scrutiny and robust defenses to ensure the safe deployment of VLMs.

---

### Official Review · Reviewer_BULD · 2024-07-11

**Soundness:** 3
**Presentation:** 2
**Contribution:** 2
**Rating:** 5
**Confidence:** 3

**Summary:**

The paper introduces a subtle attack on vision-language models (VLMs). By stealthily modifying training data, the attack influences model outputs without obvious signs of tampering. Extensive experiments demonstrate the attack's effectiveness and stealthiness, revealing significant vulnerabilities in VLMs. The findings underscore the need for improved security measures to protect against such sophisticated data poisoning techniques.

**Strengths:**

1.The paper introduces a novel clean-label attack on vision-language models (VLMs), utilizing stealthy data poisoning to manipulate multimodal outputs while maintaining a high degree of concealment.

2.The proposed method is versatile, encompassing both label attacks and persuasion attacks, demonstrating its broad applicability.

3.The approach is well-articulated, supported by comprehensive experimental evaluations, including both automated and human assessments, which validate the attack's effectiveness.

**Weaknesses:**

1.The experiments only evaluate a Llava VLM model, one dataset, and four attack tasks. Expanding the scope to include more diverse models, datasets, and attack tasks would enhance the generalizability of the findings.

2.The impact of the PGD perturbation step size on the experimental results is not discussed. Providing a range of ablation studies would help in understanding the method's effectiveness more comprehensively.

3.The performance of ShadowCast against SOTA backdoor defenses is not explored. While the poisoned dataset is imperceptible to the human, its robustness against SOTA data anomaly detection techniques needs to be assessed.

**Questions:**

1.How does the ShadowCast attack perform on VLMs trained on significantly larger and more diverse datasets?

2.Can the attack be adapted to target other types of multimodal models, such as those combining text with audio or other data modalities?

3.What are the implications of the attack on models deployed in high-stakes environments, such as healthcare or autonomous driving?

4.Are there specific characteristics of the poisoned data samples that make them more or less detectable by standard data validation processes?

**Limitations:**

1.Scalability: How the attack scales with the size of the dataset and the complexity of the model.

2.Real-world Deployment: Potential challenges and implications of deploying the attack in real-world settings.

---

> ### Author Rebuttal · Authors · 2024-08-06
>
> We thank Reviewer BUID for the detailed feedback. We are encouraged that the reviewer finds our approach novel and versatile with broad application, our method well-articulated and our experiments comprehensive. Below we address the reviewer's concerns in detail.
>
> ---
> >Weakness 1: The experiments only evaluate a Llava VLM model, one dataset, and four attack tasks...
>
> >Question 1: How does the ShadowCast attack perform ... larger and more diverse datasets?
>
> **Model**: It is not true that we "only evaluate a Llava VLM model". In the paper we did actually evaluate Shadowcast with another VLM MiniGPT4-v2-7B, as detailed in Section 4.1. Figures 11 and 12 in Appendix B.2 show effective results: the attack success rate exceeds 0.8 in the grey-box setting and reaches up to 0.6 in the black-box setting, demonstrating strong performance across VLM models.
>
>
> **Dataset**: As requested, we conduct additional experiments on another dataset ScienceQA. We evaluated Trump-to-Biden (Label Attack) and JunkFood-to-HealthyFood (Persuasion Attack) on LLaVA-1.5. The attack success rate is shown below.
>
> Trump-to-Biden (Label Attack):
> | Poison rate | 0 (clean) | 0.285% | 0.571% | 0.857% | 1.43% | 2.85% | 5.71% |
> |-|-|-|-|-|-|-|-|
> |Attack Success Rate (ScienceQA)|0|0.36|0.80|0.94|0.98|0.97|0.99|
>
> JunkFood-to-HealthyFood (Persuasion Attack):
> | Poison rate | 0 (clean) | 0.285% | 0.571% | 0.857% | 1.43% | 2.85% | 5.71% |
> |-|-|-|-|-|-|-|-|
> |Attack Success Rate  (ScienceQA)|0.05|0.16|0.19|0.65|0.77|0.95|0.96|
>
>
> We can see that poison samples crafted by Shadowcast have **similarly strong performance** when the clean training dataset changes between cc_sbu_align (in our paper) and ScienceQA. This is not surprising, given that Shadowcast works by associating image features of the original concept (like Trump) to texts from the destination concepts (like Biden). Such mechanism is independent of the clean dataset.
>
> **Attack Task**: As requested, we extended our evaluation to a more common concept pair, apple-to-banana. The attack success rate results below show that **Shadowcast is still effective for more common concept pairs** like apple-to-banana. We have put this additional experiment in our paper.
>
> | Poison rate | 0 (clean) | 0.285% | 0.571% | 0.857% | 1.43% | 2.85% | 5.71% |
> |-|-|-|-|-|-|-|-|
> |Attack Success Rate|0|0.02|0.41|0.73|0.90|0.95|0.96|
>
> ----
> >Weakness 2: The impact of the PGD perturbation step size on the experimental results is not discussed.
>
> Thank you for the suggestion! We conducted a hyper-parameter search on the initial step size and number of PGD steps. The results for the Trump-to-Biden task show that our chosen setup (init step size = 0.2/255, steps = 2000) achieves the best overall performance. This ablation study has been included for all attack tasks in the paper.
>
> | step size; step number | clean | 0.285% | 0.571% | 0.857% | 1.43% | 2.85% | 5.71% |
> |-|-|-|-|-|-|-|-|
> |0.2/255; 2000|0|0.40|0.82|0.92|0.97|0.97|0.99|
> |0.1/255; 2000|0|0.34|0.72|0.86|0.90|0.94|0.98|
> |0.1/255; 4000|0|0.37|0.80|0.92|0.96|0.97|0.99|
> |0.4/255; 2000|0|0.32|0.74|0.82|0.90|0.91|0.93|
>
> ----
> >Weakness 3: The performance of ShadowCast against SOTA backdoor defenses is not explored.
>
> As requested, we evaluated Shadowcast against DiffPure (Nie et al, ICML 2022), a **SOTA defense** that purifies poisoned images using diffusion models. We show the results on the Trump-to-Biden task below, where **Shadowcast remains effective against DiffPure**, particularly at higher poison rates. This finding is consistent across other tasks. We have updated the paper to include this result.
>
> | Poison rate | 0 (clean) | 0.285% | 0.571% | 0.857% | 1.43% | 2.85% | 5.71% |
> |-|-|-|-|-|-|-|-|
> |No defense|0|0.40|0.82|0.92|0.97|0.97|0.99|
> |DiffPure defense|0|0.06|0.14|0.23|0.69|0.91|0.97|
>
> Our results show that **neither simple measures like JPEG and image augmentation nor advanced defenses like DiffPure can effectively counter Shadowcast**, highlighting the need for more robust data sanitization methods to protect against VLM data poisoning.
>
> ----
> >Question 2: Can the attack be adapted to target other types of multimodal models, such as those combining text with audio or other data modalities?
>
> Yes! Our stealthy attack framework can be adapted to other multimodal LLMs with continuous input (e.g., audio, video), which allows for imperceptible input perturbations.
>
> ----
> >Question 3: ... implications of the attack on models deployed in high-stakes environments, such as healthcare or autonomous driving?
>
> >Limitation 2: ... challenges and implications of deploying the attack in real-world settings.
>
> In healthcare, a VLM (e.g., for analyzing patient data like CT scan) could be poisoned to recommend specific drugs, potentially benefiting some drug company. In autonomous driving, the model could be manipulated to misinterpret road signs, leading to dangerous driving decisions.
>
> ----
> >Question 4: Are there specific characteristics of the poisoned data samples that make them more or less detectable by standard data validation processes?
>
> Poisoned samples from Shadowcast are visually indistinguishable from benign ones, making them difficult to detect with standard data validation. This highlights the need for enhanced data validation techniques, as emphasized in our work.
>
> ----
> >Limitation 1: Scalability: How the attack scales with the size of the dataset and the complexity of the model.
>
> We demonstrated Shadowcast on widely-used visual instruction tuning datasets (CC, ScienceQA) and popular VLMs (LLaVA, MiniGPT), making a strong case for its applicability. Given its efficiency (less than 90 seconds per sample on the LLaVA model) and effectiveness, we expect it to scale to other datasets and models as well.
>
> ---
> Thank you again for your time and effort in reviewing our paper! Please let us know if the above explanations do not address your concerns. We are happy to answer any further questions.

---

> ### Author Response · Authors · 2024-08-10
> **Thank you for the effort in reviewing our paper**
>
> We sincerely thank the reviewer for your time and effort in reviewing our paper! We have addressed the specific points you raised regarding additional clean datasets and attack tasks, ablation on PGD step sizes, robustness against SOTA defense and other clarifications throughout our rebuttal.
>
> Please let us know whether we have fully addressed your concerns. We are more than happy to provide additional clarifications if you have further questions. Thank you!

---

> > ### Author Response · Authors · 2024-08-13
> > **Thank you for the effort in reviewing our paper!**
> >
> > We sincerely thank the reviewer for your time and effort in reviewing our paper! We have addressed the specific points you raised regarding additional clean datasets and attack tasks, ablation on PGD step sizes, robustness against SOTA defense and other clarifications throughout our rebuttal.
> >
> > Please let us know whether we have fully addressed your concerns. We are more than happy to provide additional clarifications if you have further questions. Thank you!

---

> > > ### Comment · Reviewer_BULD · 2024-08-13
> > >
> > > The authors' detailed rebuttal and additional experiments have satisfactorily addressed most of my concerns, particularly regarding the scope of datasets and tasks evaluated. However, I still have some reservations regarding the real-world applicability of the attack in diverse multimodal environments. Nonetheless, the overall contribution and technical merit lead me to raise my score to 5.

---

> > > > ### Author Response · Authors · 2024-08-13
> > > > **Thank you for your response**
> > > >
> > > > Thank you for your response! We are glad that most of your concerns have been addressed. Regarding the "real-world applicability of our attack", we emphasize that our approach is practical both in terms of the attack objective and the threat model.
> > > >
> > > > **Practicality of the Attack Objective**: In healthcare, for instance, a benign user might photograph a drug bottle for **Disease A** and ask a VLM for information. Using Shadowcast, an attacker could poison the VLM so it **incorrectly** states the drug treats a **different disease B**. To create poison samples for this through Shadowcast, the attacker would only need to pair perturbed images of Disease B drugs with corresponding texts. This demonstrates how Shadowcast can spread misinformation and cause **real-world harm**, highlighting the critical need for awareness and defenses against such attacks.
> > > >
> > > > **Practicality of the Threat Model**. It has been shown that poisoning web-scale training datasets is practical[1]. For example, one can poison 0.01% of LAION-400M for just $60 USD and also easily inject malicious examples to Wikipedia [1]. For VLMs, the training images are often chosen from these large-scale web-sourced dataset like LAION, and our proposed Shadowcast applies. In fact, the cc-sbu-align dataset we use in our paper (and also in MiniGPT4) is constructed from the Conceptual Captions dataset, which is also a web-sourced dataset and thus can be poisoned in practice. Also, the attacker can directly upload the poison images and corresponding texts (such as hash tag, short titles in the metadata) online and wait for them to be scraped for training data by, e.g., AI companies.
> > > >
> > > > In addition, Shadowcast enables stealthy poisoning to avoid human detection, which makes it more practical in real-world applications.
> > > >
> > > > [1] Carlini, Nicholas, et al. "Poisoning Web-Scale Training Datasets is Practical." 2024 IEEE Symposium on Security and Privacy (SP). IEEE Computer Society, 2024.

---

### Official Review · Reviewer_Tzvw · 2024-07-13

**Soundness:** 3
**Presentation:** 3
**Contribution:** 2
**Rating:** 5
**Confidence:** 3

**Summary:**

This paper explores the data poisoning attacks against vision language model. The Shadowcast introduces two types of attack: 1) Label Attack: the VLM generates text that misclassifies, e.g., change “Trump” to “Biden”; 2) Persuasion Attack: the VLM generates “rational” but wrong fact, e.g., illustrate the picture of “junk food” with “healthy food” description. To achieve the data poisoning, the author crafts the poisoned images inspired by clean-label poisoning; and crafts the poisoned text by replacing the text label (Label Attack) or description (Persuasion Attack).

**Strengths:**

1.Investigating the vulnerability of VLM is an interesting and active research area.

2.Involving human evaluation is not significant, however encouraged evaluation way.

3.The paper is well-written and well-organized.

**Weaknesses:**

1.Poisoning attacks on NLP/CV are well known. This paper is a bit incremental, seems to apply the data poisoning on VLMs. The way to craft poisoned samples are not technique significant: 1) For text, a separate VLM will first generate the image description; then the author replaces the class label (e.g., change “Trump” to “Biden”) or modifies the description (e.g., ask to describe the junk food as healthy food) by specific paraphrasing instructions. 2) For images, the authors leverage similar feature collision strategy in (Shafahi et al. [2018], Zhu et al. [2019]).
In this way, the contribution of this work is trivial.

2.The author evaluates the attack effectiveness on two VQA dataset. Are there other data/tasks to indicate the attack efficiency of Shadowcast?

3.There are only limited Label/Persuasive attack examples. E.g., “Trump” to “Biden”. Are there other showcases?

**Questions:**

1.What is the training details to get a well-trained backdoored VLM? E.g., how many epoch for two datasets?

2.I assume the data poisoning happens during instruction tuning stage. Does the author explore anything regarding data poisoning during pre-training?

---

> ### Author Rebuttal · Authors · 2024-08-06
>
> We thank Reviewer Tzvw for the detailed feedback. We are encouraged that the reviewer finds our paper well-written, our research interesting and timely and appreciates our human evaluation. Below we address the reviewer's concerns in detail.
>
> ---
> >Weakness 1: Poisoning attacks on NLP/CV are well known. This paper is a bit incremental, seems to apply the data poisoning on VLMs... 1) For text, a separate VLM will first generate the image description; then the author replaces the class label (e.g., change “Trump” to “Biden”) or modifies the description (e.g., ask to describe the junk food as healthy food) by specific paraphrasing instructions. 2) For images, the authors leverage similar feature collision strategy in (Shafahi et al. [2018], Zhu et al. [2019]). In this way, the contribution of this work is trivial.
>
> While our approach is simple, it tackles **novel challenges specific to poisoning VLMs**. Unlike prior attacks on image classification models (Shafahi et al. [2018], Zhu et al. [2019]), VLM poisoning requires generating free-form text for poison samples. Shadowcast introduces a caption-then-refine strategy, where **paraphrasing with an LLM is critical** for maximizing the attack's potency by crafting texts that clearly emphasize the target concept. The following results for the JunkFood-to-HealthyFood (Persuasion Attack) task show that **omitting this step significantly reduces attack effectiveness**. This highlights one of the non-trivial contributions of our work.
>
>
> | Poison rate | 0 (clean) |  0.857% | 1.43% | 2.85% | 5.71% |
> |-|-|-|-|-|-|
> |with paraphrasing (ours)|0.09|0.57|0.70|0.93|0.95|
> |without paraphrasing|0.09|0.29|0.44|0.59|0.64|
>
>
> In addition, we would like to highlight the **novel attack objectives** and **practical threat models** that are **unachievable in prior attacks**:
>
> * Persuasion Attack: We introduce the persuasion attack objective, where poisoned VLMs generate coherent misinformation, subtly altering user perceptions—an attack that **cannot** be executed by prior vision system attacks like those on CLIP or image classifiers (Shafahi et al. [2018], Zhu et al. [2019]).
>
> * Stealthy Poisoning: Our work confirms the feasibility of stealthy poisoning against VLMs, evading human detection—a challenge **not** addressed by LLM poisoning.
>
> * Threats to Benign Users: Shadowcast poses threats to benign users with innocuous prompts, **unlike** jailbreaking attacks that rely on adversarial prompts.
>
> * Practical Scenarios: We validate Shadowcast across practical attack scenarios, demonstrating its effectiveness across architectures in **black-box settings** and its resilience to common countermeasures during training.
>
> ---
> >Weakness 2: The author evaluates the attack effectiveness on two VQA dataset. Are there other data/tasks to indicate the attack efficiency of Shadowcast?
>
> As requested, we conduct experiments on another dataset ScienceQA. Using 3,500 random samples from the train split, we evaluated Trump-to-Biden (Label Attack) and JunkFood-to-HealthyFood (Persuasion Attack) on LLaVA-1.5. The attack success rate is shown below.
>
> Trump-to-Biden (Label Attack):
> | Poison rate | 0 (clean) | 0.285% | 0.571% | 0.857% | 1.43% | 2.85% | 5.71% |
> |-|-|-|-|-|-|-|-|
> |Attack Success Rate (ScienceQA)|0|0.36|0.80|0.94|0.98|0.97|0.99|
>
> JunkFood-to-HealthyFood (Persuasion Attack):
> | Poison rate | 0 (clean) | 0.285% | 0.571% | 0.857% | 1.43% | 2.85% | 5.71% |
> |-|-|-|-|-|-|-|-|
> |Attack Success Rate  (ScienceQA)|0.05|0.16|0.19|0.65|0.77|0.95|0.96|
>
>
> We can see that poison samples crafted by Shadowcast have **similar strong performance** when the clean training dataset changes between cc_sbu_align (in our paper) and ScienceQA. This is not surprising, given that Shadowcast works by associating image features of the original concept (like Trump) to texts from the destination concepts (like Biden). Such mechanism is independent of the clean dataset.
>
> ---
> >Weakness 3: There are only limited Label/Persuasive attack examples. E.g., “Trump” to “Biden”. Are there other showcases?
>
> As requested, we consider another attack task where the attacker tries to poison the VLM to predict "banana" when prompted with "apple" images. We choose apple-banana concept pair as additional experiments since they are more common concept than Biden-Trump (used in the paper). The attack success rate results are shown below.
>
> | Poison rate | 0 (clean) | 0.285% | 0.571% | 0.857% | 1.43% | 2.85% | 5.71% |
> |-|-|-|-|-|-|-|-|
> |Attack Success Rate|0|0.02|0.41|0.73|0.90|0.95|0.96|
>
> The results above show that **Shadowcast is still effective for more common concept pairs** like apple-to-banana. Comparing with the results of Trump-to-Biden and EngineLight-to-LowFuelLight tasks in Figure 3 (page 7), the attack success rate results of apple-to-banana task are very similar. We will put this additional experiment in our paper.
>
> ---
> >Question 1: What is the training details to get a well-trained backdoored VLM? E.g., how many epoch for two datasets?
>
> As noted in line 204, section 4.1, the LLaVA-1.5 model is trained for one epoch on the poisoned dataset. Detailed training information for both LLaVA and MiniGPT4-v2 models can be found in lines 199-208 of section 4.1 and Appendix B.3.
>
> ---
> >Question 2: I assume the data poisoning happens during instruction tuning stage. Does the author explore anything regarding data poisoning during pre-training?
>
> Poisoning pre-training data is an interesting direction, but due to computational constraints, we focus on instruction tuning. Instruction tuning requires much less data and therefore it demands stealthier attacks to evade manual inspection, making it more technically challenging. Our work addresses these challenges, demonstrating effective stealthy data poisoning in this critical stage.
>
>
> ---
> Thank you again for your time and effort in reviewing our paper! Please let us know if the above explanations do not address your concerns. We are happy to answer any further questions.

---

> > ### Comment · Reviewer_Tzvw · 2024-08-12
> >
> > Thanks the author's rebuttal. I still keep my attitude regarding the "This paper is a bit incremental, seems to apply the data poisoning on VLMs."
> >
> > The author claims "a novel challenge of creating free-form texts for poison samples.", where **paraphrasing with an LLM is critical**. This paraphrasing is the prompt engineering utilizing other LLM/VLMs, e.g., crafting the poisoned samples, for example: replacing the biden to trump, or changing the description of junk food with the description of healthy food. I do not think this is significant.
> >
> > However, the author did address some of my concerns, thus I will increase my score to 5.

---

> > > ### Author Response · Authors · 2024-08-12
> > > **Thank you for the response**
> > >
> > > Thank you for the positive feedback!
> > >
> > > We would like to emphasize that "applying data poisoning to VLMs" presents a unique contribution to AI security.
> > > This approach uniquely enables stealthy attacks on benign prompts while spreading free-form misleading information—capabilities that are **not possible in other settings**.
> > > Exposing these new and practical vulnerabilities in VLMs, which are increasingly used in various applications, is crucial for alerting the community and advocating for stronger data scrutiny and robust defenses to ensure the safe deployment of VLMs.

---

> ### Author Response · Authors · 2024-08-10
> **Thank you for the effort in reviewing our paper!**
>
> We sincerely thank the reviewer for your time and effort in reviewing our paper! We have addressed the specific points you raised regarding novelty, additional clean datasets and attack tasks and other clarifications throughout our rebuttal. We also evaluate Shadowcast against SOTA defense methods and observe promising results, which is provided in the global response.
>
> Please let us know whether we have fully addressed your concerns. We are more than happy to provide additional clarifications if you have further questions. Thank you!

---

### Author Rebuttal · Authors · 2024-08-06

We thank all reviewers for their valuable feedback. We are pleased that the reviewers found our proposed approach and attack objective novel (BULD, RqR6), and recognized our paper as well-written (Tzvw, BULD, u5mc, RqR6). The reviewers also considered our work interesting and timely (Tzvw, u5mc), appreciated our human evaluation (Tzvw), and commended the comprehensive and strong experiments we conducted (BULD, u5mc, RqR6).

Below, we summarize our response to common concerns and misunderstanding as well as additional experiment results.

---

### Concern 1: Novelty

While some reviewers (BULD, RqR6) recognized the novelty of our attack objective and approach, others (Tzvw, u5mc) expressed concerns about the simplicity of Shadowcast. We emphasize that simplicity does not undermine the novelty or effectiveness of Shadowcast. Shadowcast introduces novel attack objectives and practical threat models unachievable in prior works and also deals with novel challenges of effectively creating free-form texts for poison samples.

First, our approach, while simple, highlights the **novel attack objectives** and **practical threat models** that are **unachievable in prior attacks**:

* Persuasion Attack: We introduce the persuasion attack objective, where poisoned VLMs generate coherent misinformation, subtly altering user perceptions—an attack that **cannot** be executed by prior vision system attacks like those on CLIP or image classifiers.

* Stealthy Poisoning: Our work confirms the feasibility of stealthy poisoning against VLMs, evading human detection—a challenge **not** addressed by LLM poisoning.

* Threats to Benign Users: Shadowcast poses threats to benign users with innocuous prompts, **unlike** jailbreaking attacks that rely on adversarial prompts.

* Practical Scenarios: We validate Shadowcast across practical attack scenarios, demonstrating its effectiveness across architectures in **black-box settings** and its resilience to common countermeasures during training.

Second, unlike prior poisoning attacks on image classification models, poisoning VLMs requires a **novel challenge of creating free-form texts for poison samples**. Shadowcast addresses it through a caption-then-refine strategy described in Section 3.3, where **paraphrasing with an LLM is critical** for maximizing the attack's potency by crafting texts that clearly emphasize the target concept. The following results for the JunkFood-to-HealthyFood (Persuasion Attack) task show that **omitting this step significantly reduces attack effectiveness**. This highlights one of the non-trivial contributions of our work.

| Poison rate | 0 (clean) |  0.857% | 1.43% | 2.85% | 5.71% |
|-|-|-|-|-|-|
|with paraphrasing (ours)|0.09|0.57|0.70|0.93|0.95|
|without paraphrasing|0.09|0.29|0.44|0.59|0.64|

---

### Concern 2: State-of-the-art Defense

Reviewer Tzvw and BUID suggest evaluating Shadowcast under SOTA defense methods. As requested, we evaluated Shadowcast against DiffPure (Nie et al, ICML 2022), a **SOTA defense** that purifies poisoned images using diffusion models. We show the results on the Trump-to-Biden task below, where **Shadowcast remains effective against DiffPure**, particularly at higher poison rates. This finding is consistent across other tasks. We have updated the paper to include this result.

| Poison rate | 0 (clean) | 0.285% | 0.571% | 0.857% | 1.43% | 2.85% | 5.71% |
|-|-|-|-|-|-|-|-|
|No defense|0|0.40|0.82|0.92|0.97|0.97|0.99|
|DiffPure defense|0|0.06|0.14|0.23|0.69|0.91|0.97|

This result, along with results in our submitted version, show that **neither simple measures like JPEG and image augmentation nor advanced defenses like DiffPure can effectively counter Shadowcast**, highlighting the need for more robust data sanitization methods to protect against VLM data poisoning.

---

### Concern 3: More clean datasets and attack tasks

Reviewer Tzvw and BUID suggest evaluating on more clean datasets and attack tasks.

**Clean Dataset**: As requested, we conduct additional experiments on another dataset ScienceQA. We evaluated Trump-to-Biden (Label Attack) and JunkFood-to-HealthyFood (Persuasion Attack) on LLaVA-1.5. The attack success rate results below show **similarly Shadowcast's strong performance** when the clean training dataset changes between cc_sbu_align (in our paper) and ScienceQA.

Trump-to-Biden:
| Poison rate | 0 (clean) | 0.285% | 0.571% | 0.857% | 1.43% | 2.85% | 5.71% |
|-|-|-|-|-|-|-|-|
|Attack Success Rate (ScienceQA)|0|0.36|0.80|0.94|0.98|0.97|0.99|

JunkFood-to-HealthyFood:
| Poison rate | 0 (clean) | 0.285% | 0.571% | 0.857% | 1.43% | 2.85% | 5.71% |
|-|-|-|-|-|-|-|-|
|Attack Success Rate  (ScienceQA)|0.05|0.16|0.19|0.65|0.77|0.95|0.96|


**Attack Tasks**: As requested, we extended our evaluation to a more common concept pair, apple-to-banana. The attack success rate results below show that **Shadowcast is still effective for more common concept pairs** like apple-to-banana.

| Poison rate | 0 (clean) | 0.285% | 0.571% | 0.857% | 1.43% | 2.85% | 5.71% |
|-|-|-|-|-|-|-|-|
|Attack Success Rate|0|0.02|0.41|0.73|0.90|0.95|0.96|

----
### Paper updates

* (Tzvw, BUID) Additional experiments on SOTA defense DiffPure in experiment section 4.5.
* (RqR6) Correction on the text descriptions in line 375.
* (Tzvw, BUID) Additional experiments on more clean datasets (ScienceQA) and attack tasks (apple-to-banana) in Appendix B.
* (BUID) Ablation studies on PGD step size in Appendix B.
* (u5mc) Adding a non-stealthy baseline in Appendix B.

---
We thank all reviewers again for the time and effort in reviewing our paper! Please let us know if the above explanations do not address your concerns. We are happy to answer any further questions.

---

### Decision · Program_Chairs · 2024-09-25

**Decision:**

Accept (poster)

**Comment:**

All reviewers have provided positive scores for this submission, highlighting its strengths in novelty, well-written manuscript, and extensive experiments. The AC carefully reviewed the paper and concurred with the reviewers' assessments, therefore supporting the decision to accept this submission.